# Item Analysis of the Czech Version of the WJ IV COG Battery from a Group of Romani Children

**DOI:** 10.3390/ijerph181910518

**Published:** 2021-10-07

**Authors:** Alena Kajanová, Tomáš Urbánek, Tomáš Mrhálek, Stanislav Ondrášek, Olga Shivairová, Jan Hynek

**Affiliations:** 1Faculty of Health and Social Studies, University of South Bohemia in České Budějovice, 37005 České Budějovice, Czech Republic; ondrasek@zsf.jcu.cz (S.O.); oshivairova@zsf.jcu.cz (O.S.); 2Institute of Psychology, The Czech Academy of Sciences, 61662 Brno, Czech Republic; tour@psu.cas.cz; 3Department of Pedagogy and Psychology, Faculty of Education, University of South Bohemia in České Budějovice, 37005 České Budějovice, Czech Republic; tmrhalek@gmail.com (T.M.); hynekj00@pf.jcu.cz (J.H.)

**Keywords:** Romani minority, Romani children, Woodcock–Johnson IV, differential item functioning

## Abstract

The objective of the article is to present an item analysis of selected subtests of the Czech version of the WJ IV COG battery from a group of Romani children, ages 7–11. The research sample consisted of 400 school-aged Romani children from the Czech Republic who were selected by quota sampling. A partial comparative sample for the analysis was the Czech population collected as norms of the Czech edition of © Propsyco (*n* = 936). The Woodcock–Johnson IV COG was used as a research tool. Statistical analysis was performed in Winstep software using Differential Item Functioning; differences between groups were expressed in logits and tested via the Rasch–Welch *T*-test. It was discovered that higher item difficulty was noted in the verbal subtests, although variability in item difficulty was found across all subtests. The analysis of individual items makes it possible to discover which tasks are most culturally influenced.

## 1. Introduction

Many tests that measure cognitive ability are developed and standardized on the majority population. Since the beginning of intelligence testing to quantify measures of intellectual ability and mental level, several questions have arisen regarding the validity of intelligence tests to measure executive function. Michell [1] casts doubt on the common practice of psychological measurement of cognitive function. Many intelligence tests result in a “number” that says nothing about the structure of a trait that is not internally homogeneous. Reynolds [2] argues that tests are closely related to the culture in which they are developed; thus, they are culturally loaded, resulting in differential performance among individuals who have different racial–ethnic backgrounds, which often leads to unequal treatment of minorities. The need to take cultural differences into account during cognitive testing is mentioned by Hambleton and Zenisky [3] Many cognitive ability tests fail to identify a variety of important characteristics and do not include relevant questions from the perspective of minority populations. For this reason, it is important to apply a correct methodological approach, which can only be achieved by correctly translating and adapting the test directly to the minority population.

The paper aims to present an item analysis of the basic subtests of the Czech version of the WJ IV COG battery for a group of Romani children aged 7–11 years in relation to the normative Czech population. The results provide information on the suitability of the WJ IV test for the study population and will therefore verify the potential of using the tool to measure the non-majority population. A detailed presentation of items with different difficulties can provide important information for the users of this tool.

## 2. Theoretical Foundations

According to qualified estimates, there are approximately 250,000 Roma (Government of the Czech Republic). Ref. [4] living in the Czech Republic (CZ), approximately 150,000 of whom live in socially excluded communities, which we will address in the text, and of which approximately 80% are Roma (Čada et al.) [5]. The Roma minority in the Czech Republic face a double disadvantage, i.e., ethnicity and socioeconomic status (SES) (Kozubík et al) [6].

The main criticisms of cognitive ability tests used for social and ethnic minorities are based on the fact that most cognitive ability tests are developed for European–American or “Western” civilizations and do not reflect the specificities of other cultures (Thaler et al.) [7]. The construct validity of these tests does not sufficiently reflect the specificities of other cultures. Unless test characteristics are related to the construct and to all groups on which the test is used, there is a risk that the test will measure different characteristics in members of cultures, i.e., characteristics other than those it was initially intended to measure. Thaler et al. [7] further analyzed possible variables affecting construct validity related to minorities. One of the variables is the so-called “differential item functioning” and reflects the probability of responding incorrectly to a test item as a function of membership in a particular social minority. This differential item can be “uniform,” meaning that a given social minority group gives incorrect answers to all items across the test, or “specific,” where a given group answers differently, but only on specific test items. Therefore, standardization of cognitive ability tests needs to be carried out on the whole population, and item analysis of the tests needs to examine different responses not only quantitatively but also qualitatively.

The cultural load of cognitive ability tests was addressed by Snyderman and Rothman [8] in the 1980s; in addition to cultural background, they examined other factors affecting performance on these tests. They concluded that in terms of the importance of intervening factors, the quality of education and cultural determinants closely related to socioeconomic status were the leading factors. This was confirmed by Nisbett et al. [9] when they stated that the differential performance of minority groups on these tests is more likely attributable to social and environmental factors.

Originally, it was thought that subtests with a large verbal component were, in particular, subject to cultural bias, while tests with non-verbal components, such as those related to mathematical ability and concept formation, were not subject to cultural bias. Thaler et al. [7] reported a review study in which they analyzed the results of the WAIS-III for measuring cognitive ability. The research demonstrated that Hispanic and African American minorities perform lower on the verbal subtests of the WAIS-III and across subtests, including arithmetic and spatial reasoning. Therefore, all subtests must be considered as potentially subject to cultural bias. Dočkal and Filípková [10] found similar results using a sample of Romani children. Ferjenčík, Slavkovská, and Kresila [11] stressed the need to develop specific standards for testing the cognitive abilities of Romani children, especially those from culturally disadvantaged backgrounds.

The question of the legitimacy of cognitive ability tests for Romani children from culturally disadvantaged backgrounds was addressed by Gaertner and Tellegen [12] They identified factors likely to influence cognitive ability tests in these children, e.g., more frequent absenteeism, leaving school early, language barriers, motivation, concentration, and family situation, the latter of which mainly refers to parental value orientations and lack of parental support for education-related activities.

Dolean et al. [13] reported that socially disadvantaged backgrounds and low socioeconomic status negatively affect performance even on ability tests that are not expected to be significantly influenced by culture (e.g., Raven’s Coloured Progressive Matrices). Engelhardt, Church, Harden, and Tucker-Drob [14] found that socioeconomic status was a significant factor influencing performance on mathematical and nonverbal subtests (as measured using the WAIS-II). Similar findings were reported in a study by Piccolo et al. [15], who found that performance on language, memory, and executive functions varied relative to socioeconomic status. Crane [16] looked at the influence of several factors that may affect the mathematical performance of children aged 5–9 years. He included family background and socioeconomic status. Factors such as ethnicity did not have an effect in terms of incremental variance.

Weiss and Soklofske [17] questioned the validity of using ethnicity as an independent variable to explain the difference in cognitive ability since socioeconomic status is much more reflective of variance in IQ scores than ethnicity. Valencia and Suzuki [18] made a similar argument. Rindermann, Becker, and Coyle [19] stress the need to take into account the interdependence of these factors, whereby initially low socioeconomic status creates reduced opportunities for obtaining a quality education, and the consequence is again low socioeconomic status, thereby exacerbating the differences in cognitive ability between minority and majority populations. Denglerová [20] also points out that David Wechsler’s original definition of intelligence as “an individual’s complex global capacity to think rationally, cope successfully with the demands of the environment, understand the world, and cope effectively with challenges…” explicitly mentions the relationship between intelligence and an individual’s background, which is why most intelligence tests are not appropriate for people who are not part of the majority society. Denglerová’s [21] research focused on comparing ways of categorizing Romani children from socially disadvantaged backgrounds and children from the majority society. Romani children categorized objects according to a different key than children from the majority society. According to Denglerová [20], the correct approach to testing the intelligence and cognitive abilities of children from culturally disadvantaged backgrounds is a constructivist approach that considers the context of the native culture and cultural background.

Ferjenčík [11] also points to the inappropriateness of using a generalized approach for assessing children’s cognitive abilities from culturally disadvantaged backgrounds. Most tests of intelligence and cognitive ability were developed and standardized on the majority population; to be useful for assessing children from culturally disadvantaged backgrounds, these tests need to undergo a thorough adaptation process, and special standards need to be developed.

## 3. Methodology

### 3.1. Design

The research was part of the process of normalizing the WJ IV test for the Czech population. It was quota research using selected items from the cognitive ability test, individually administered to a sample of Roma primary school pupils.

### 3.2. Participants

The research sample consisted of 400 young school-age Romani children from all administrative parts of the Czech Republic, selected by quota by age (two age categories were represented: 7–8 years and 10–11 years), gender, and social exclusion expressed as residence in socially excluded communities (Chad) [22], so half of the respondents lived in socially excluded communities, and half lived in non-excluded communities (this variable is further analyzed in another article). The condition for inclusion in the research was self-identification by the participant’s families with the Roma ethnic group. The research was carried out in all districts of the Czech Republic, and recruitment took place through a gatekeeper; snowball sampling was also used. The project was approved by the Ethics Committee of the University of South Bohemia in České Budějovice, Faculty of Health and Social Studies, nr. 12062018. The participants and their legal representatives were acquainted with the aim and purpose of the research. Data were collected anonymously and did not contain personal information.

### 3.3. Methods

The Woodcock–Johnson IV test (McGrew, LaForte, and Schrank) [23] was used as the research instrument. The test administration was performed by trained psychologists who completed the relevant courses for users of the tool. Information about the subtests used, their structure, and test content are described in more detail in the Results section. Individual items were scored as correct or wrong (i.e., 0–1 points).

### 3.4. Measurement

The Item Response Theory (IRT) paradigm was used; namely, the Rasch model evaluated item difficulty on a multi-item scale to control for overall scores on the subtest. IRT evaluates item difficulty for dichotomous items. The probability of a specified response (right/wrong answer) is modeled as a function of the person and item parameters (0.0 is the average item difficulty for each subtest; the more positive the value, the more difficult the item, while negative numbers indicate easier items). Differential item functioning (DIF) methods are procedures to detect items that are functioning differently across groups of individuals (de Ayala) [24] For analysis, we used a procedure developed for the WJ IV test battery for the US population. Because this research only focuses on children, we describe the results for the first twenty items in each subtest, which are most relevant for testing respondents in the given age groups. Norms from the Czech population (Czech edition of © Propsyco (*n* = 936)) were used as a comparative set for item difficulty in our analysis.

### 3.5. Analysis 

Statistical analysis was performed in Winstep software (Linacre) [25] using Differential Item Functioning contrast (DIF contrast), which calculates the difference in difficulty (reported as DIF measure) of an item between two groups. The DIF contrast is calculated as the difference between the two DIF measures in logits. The Czech standardization norms were used as a baseline, and DIF was computed pairwise relative to those groups. Differences in item difficulty were tested using the Rasch–Welsch *T*-test (*p* < 0.05). The results presented the significant different item with the criterion of DIF contrast level greater than 0.5 logit (mentioned by Linacre [25] as minimally noticeable; logit > 0.5 was used as a criterium in the original standardization (McGrew, LaForte, and Schrank [23].

## 4. Results

The first test, 1T Oral vocabulary, which consists of two subtests, 1A Synonyms, and 1B Antonyms, uses word manipulation and language in general to examine vocabulary level. In the test, the participant uses a cognitive process based on semantic activation, access, and attribution and responds to auditory stimuli. The presented results are supported by the focus of the T1 test on specific CHC abilities (Cattell–Horn–Carroll), i.e., comprehension–knowledge (Gc), lexical knowledge (VL), and language development (LD) (=VL/LD Vocabulary).

A difference was identified in subtest 1 Vocabulary on Task A Synonyms (Rasch–Welch *p* < 0.05, DIF contrast logit > 0.5). For Romani, the DIF measure was lower for seven items (automobile, beautiful, small, conceal, quiet, shine, and nap), whereas for the majority population, it was nine items (stare, giggle, devour, palm, part, obvious, capitulate, ruin, and plague). Significant differences towards lower difficulty (DIF contrast logit > 1) were found for Romani children on four items and the majority children on six items. Regarding subtest 1B, antonyms, the words brother, cry, silent, cheap, later were easier for Roma. The antonyms from item 11 onwards were significantly more difficult for Romani children. DIF contrast logit > 1 was only recorded for the item “brother.” For the majority children, the simpler words were: in front, target, buy, build, optimistic, and later. A DIF contrast logit > 1 was recorded for the following items: in front, build, and optimistic.

The T2 Number series test works on the principle of determining a number sequence, which involves the representation and manipulation of points on a number axis. The identification of the principle and its application to complete the series is essential. This test works with a focus on fluid intelligence (Gf—fluid reasoning), induction (I—induction), and quantitative reasoning (RQ—quantitative reasoning).

In subtest 2, a significant difference was found between groups for ten items. For the Romani group, the DIF contrast was found to be lower for five items, and for the majority group, it was five items, of which a significant DIF contrast (i.e., logit > 1) was found for three items for the Romani and two items for the majority group.

Interpretively, the increase in DIF contrast starting from item 13 was important, where the change of assignment to descending series and multiples begins. For Romani children, the increase in the task’s difficulty resulted in a deterioration compared to the majority population, and the items were more difficult for them, but as the task type continued, Romani children adapted to this type of task, and the error rate decreased.

The T3 Verbal Attention test focuses on cognitive processes based on controlled executive functions, working memory capacity, recoding of acoustic verbalized stimuli held in immediate awareness, selective auditory attention, and attentional control. It uses the CHC capabilities of short-term working memory (Gwm), working memory capacity (WM), and attentional control (AC).

In subtest 3, there was a significant difference (*p* < 0.05 and DIF logit > 0.5) between groups on 15 items. Nine items with higher DIF measures and six items with lower DIF measures were found for the Romani group. A significant difference (logit > 1) in favor of the Romani group was found for only one item, while for the majority group, there were five items, of which two items showed DIF contrast logit > 2.

In the T5 Phonological Processing test, we dealt with two parts, namely, 5A Word Access and 5C Substitution. The test works with CHC abilities: Ga—auditory processing, PC—phonetic encoding, Glr-FW; long-term retrieval—word fluency, Glr-LA; long-term retrieval—speed of lexical access. It focuses on cognitive processes based on semantic activation and lexical access, and speed. The test is designed to offer a comprehensive view of auditory processing, which is crucial in the development of language and general cognitive abilities (Anthony) [26].

In subtest 5A, a significant difference was found for eight items. Of these, five items were more difficult for Romani and three for the majority children. Two items that were easier for Romani and one for the majority children showed a contrast greater than one logit. The items that were more difficult for Romani children were specific in that they required the listing of words that had a particular vowel in the middle of the word. On the other hand, items requiring the listing of words beginning with a particular consonant, which are not very frequent in Czech and thus limit the choice of possible answers, were easier for Romani.

In the case of subtest 5C, two items were easier for the Romani children, one of which had a DIF contrast greater than one logit), while five items were more difficult, all with a DIF contrast greater than one logit). For Romani, the easier items were based on the substitution of the first syllable in words, while the more difficult items were those from number 16 onwards, which required the substitution of several syllables in different places in the word.

The T6 Story Recall test focuses on the cognitive process of constructing propositional representations and recoding them. It uses CHC abilities based on long-term memory (Glr), meaningful memory (MM), processing speed (GsLS), and listening ability (Gs).

Subtest 6, based on the reproduction of stories, was more difficult for Romani for those items that contained words that are less frequent in everyday life (e.g., coast, thimble, the law of probability, etc.). The language barrier explains this difficulty since Roma children do not encounter most of these words at all.

Test 7, Visualization, is divided into two parts: T7A—Spatial Relations, and T7B—Block Rotation. The principle of this subtest is the detection of spatial ordering. Respondents are asked to identify two-dimensional elements that form a particular shape, identify spatially rotated blocks that match a template, and mentally manipulate visual objects in space. The test targets the areas of visualization (Vz) and visual processing (Gv).

In subtest 7A, four items were easier for Romani and two for the majority. In 7B, the difficulty of the items for Romani children increased as the difficulty of the items increased. Five items from the beginning of the subtest were easier for Romani, and six were more difficult (two of which had a DIF contrast higher than one logit). Roma children started to have more difficulties from item 13 onwards.

Test 9, Concept Formation, is a rule-based categorization with the principle of switching between induction and inference rules. The goal of the test is to identify, categorize, and establish rules. The test focuses on fluid reasoning (Gf) and induction (I).

For subtest 9, we recorded six items with lower difficulty for the majority children and two for the Romani (both with a DIF contrast higher than one logit). The more difficult items for Romani children were from item 11 onwards.

Test 10, Number Reversed, is aimed at listening to and memorizing a number series but in reverse order. It targets short-term working memory (gmw), working memory capacity (WM), and attentional control (AC). It assesses the extent of comprehension and recoding in working memory as well as working memory capacity and attentional capacity.

In subtest 10, two items were easier for Romani (one with a higher DIF contrast than one logit) and four for the majority children. Romani children started to have difficulty with item 13.

Subtest 11, the Number-Pattern Matching test, is based on quickly locating and circling identical numbers from a given set. It mainly tests temporal visual perception and matching and visual discrimination and division of attention. We observed processing speed (Gs) and perceptual speed (P). The 17 Pair Cancellation test aims to assess executive processing, attentional maintenance and control, inhibition, and interference. The respondent is tasked to quickly locate and label the repeated pattern. We observe processing speed (Gs), perceptual speed (P), spatial scanning (Gv-SS; visual processing—spatial scanning), and attentional control (Gwm-AC; short-term working memory—attentional control).

In the case of subtest 11, it turns out that the Romani scored better than the majority in the third minute of the task. In the case of subtest 17, Romani scored better in the first minute and worse in the third minute of the task.

## 5. Discussion

As many studies have shown (see Theoretical Foundations), it is important to examine diagnostic tools in terms of their cultural independence. As part of the development of the WJ IV, experts on minority groups in the United States, where the test was developed, were consulted (McGrew, LaForte, Schrank) [23] The normative set represented the ethnic proportionality of the US population. Creating Czech norms for the WJ IV battery was specific to the original edition in that there was no proportional representation of ethnic minority groups. For this reason, a partial study focusing on diagnostics for Romani children was carried out (the output of which is this article) since the Romani minority is the largest in the Czech Republic (Mareš, Horáková, Rákoczyová) [27].

This article deals with the validation of individual items and their differences between Romani pupils and the Czech population. Analysis using the differential item function, which allowed us to examine changes in the relative difficulty of the item for each group separately, is a suitable procedure for testing both cultural bias and cultural sensitivity of the test and provides insight into the structure of the questions, allowing for a deeper analysis and discovery of intergroup differences in proficiencies that are part of the test items. A deeper (qualitative) analysis of individual tasks was conducted based on the DIF contrast between Romani children and the norms of majority children.

Table 1 shows the differences between majority and Romani children using a differential item analysis of the different subtests. In verbal subtests, we can see a greater DIF contrast greater than one logit compared to the other subtests. This points to a greater variance in item difficulty. The greatest number of items with higher difficulty levels for Romani children was found in verbal subtest 1A (antonyms), where 30% of the examined items showed a DIF contrast greater than one logit. In T3 and T5A, DIF contrasts greater than 1 < logit was 25% of the items with lower difficulty levels for the majority children. 

In verbal subtests focused on vocabulary, we saw the greatest differences between individual groups. Subtest 1A, Synonyms, showed that a greater proportion of items were easier for the majority children, but the variance between individual items was especially significant, i.e., the fact that most items show differences between groups (45% had lower difficulty levels for majority children vs. 35% for Roma children). This result points to the crucial role of language in psychological testing. Conversely, verbal subtests that require more complex language skills have a significantly higher proportion of items that were easier for majority children (this can be seen in part in subtest 1B, Antonyms, and is fully demonstrated in subtests 5A—Word Access and 5B—Substitution). The subtests primarily based on fluid reasoning and induction did not show significant differences between groups. Interesting findings were also seen on visual ability tests, which indicate dependence on a particular type of task, as well as ambiguity in tests involving attention concentration and memory ability. These tasks should be addressed in more detail in follow-up research.

As McGrew, LaForte, and Schrank [23] point out, verbal subtests can cause complications for children who do not grow up in a family where the dominant language is the same as the test battery. We can attribute the results to bilingualism, the cultural load of the test, socioeconomic status or social exclusion, and differing item difficulty. Further analysis of verbal subtests (Mrhálek and Kajanová, in press) found specific patterns; while the items were intentionally arranged with increasing difficulty (which works well for majority children), Romani children often score better on more demanding items once they understand the principle of the task (i.e., when an item using the same principle is repeated in the test). The use of more practice items could be a good solution to identify individual abilities on tasks that require the use of procedures that are less familiar to minority children than to majority children.

There was also between-group variability in DIF measures on the nonverbal subtests, but the number of items with high DIF contrasts was lower than on the verbal subtests. Dočkal and Filípková [7] compared children from the majority population with Romani children from disadvantaged backgrounds using a test of school readiness, mainly in terms of visual differentiation, graphomotor skills, and mathematical abilities. Except for the graphomotor skills test, where performance was comparable, Romani children scored worse than non-Romani children. There was also a noticeable difference on tests of mathematical abilities; the authors interpret this difference as mathematics skills being related to the language skills of the majority population (e.g., more–less, times). Therefore, mathematical abilities cannot be completely separated from verbal abilities.

A limitation was the uneven size of the groups, i.e., majority children vs. minority children. This was solved by using an item analysis procedure, which does not require the same size groups, as can be seen in its use in developing the original American standards. We focused primarily on the usability of the analysis and recommendations for improving the Czech versions of the tests for measuring the minority population. In this study, we did not analyze the effects of SES. Comparing differences in results relative to family socioeconomic status (and factors affecting social exclusion) involves a level of complexity and requires a level of analysis that makes it part of other publication outputs for this project.

One of the strengths of this study was the way in which a minority sample was selected, which was quota-stratified, and the individual quotas were balanced. The size of the minority group is relatively large due to the size of Czech standards.

The implication of these results consists of paying attention to testing minority populations, specifically for subtests, which we point out in the article. Although the differences found may be different for different minority groups, the article touches on several important empirical principles for the construction of culturally universal cognitive ability tests.

## 6. Conclusions

This article dealt with the introduction of item analysis and compared a group of Romani children with a group of children from the majority population. Item analysis showed that differences in item difficulty were related to the overall abilities of the group; thus, examining the differences between groups can provide information about which questions on the subtests are more difficult for Romani children. This can reveal otherwise hard-to-detect cultural influences affecting the reliability of cognitive ability tests used for this minority.

Greater item difficulties were noted in the verbal subtests, although variability in item difficulty was found across all subtests. Consistent with theory, language proficiency, particularly with respect to vocabulary, appears to have the greatest impact on differences in cognitive ability tests among minority groups.

Data obtained from this project were provided to the certified publisher of the WJ IV COG tool in the Czech Republic for further use. Our item analysis can be useful for a practical analysis since it reflects cultural determinants affecting the relative difficulty of each item. This information may be useful for further work devoted to improving the reliability of this cognitive ability battery when measuring non-majority populations.

## Figures and Tables

**Table 1 ijerph-18-10518-t001:** The number of items with a different difficulty for majority children relative to Romani children.

Subtest	CHC Abilities	Lower DIF for Roma (*n* = 400)	Lower DIF for Majority (*n* = 936)
		Logit > 0.5 (Logit > 1 *)	Logit > 0.5 (Logit > 1 *)
T 1 Vocabulary A Synonyms	Comprehension–Knowledge (Gc), Lexical Knowledge (VL), and Language Development (LD)	7 (4)	9 (6)
T 1 Vocabulary B Antonyms		5 (1)	6 (3)
T 2 Number Series	Fluid reasoning (Gf), Induction (I), and Quantitative Reasoning (RQ)	5 (3)	5 (2)
T 3 Verbal Attention	Short-Term Working Memory (Gwm), Working Memory Capacity (WM), and Attentional Control (AC)	6 (1)	5 (5)
T 5 Phonological Processing A Word Access	Auditory Processing (Ga), Phonetic Encoding (PC), Long-Term Retrieval—Word Fluency (Glr-FW), and Long-Term Retrieval—Speed of Lexical Access (Glr-LA)	3 (1)	5 (2)
T 5 Phonological Processing C Substitution		2 (1)	5 (5)
T 7 Visualization A Spatial Relation	Visualization (Vz), and Visual Processing (Gv)	4 (0)	2 (0)
T 7 Visualization B Block Rotation		5 (0)	6 (2)
T 9 Concept Formation	Fluid Reasoning (Gf), and Induction (I)	2 (2)	6 (0)
T 10 Number Reversed	Short-Term Working Memory (gmw), Working Memory Capacity (WM), and Attentional Control (AC)	2 (1)	4 (0)

* Logits > 1 are a subpart of reported items logit > 0.5; reported items are significant at *p* < 0.05 (Rasch–Welch *T*-test). CHC—Cattell–Horn–Carroll; DIF—Differential item functioning.

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
