# Peer review of "Item Analysis of the Czech Version of the WJ IV COG Battery from a Group of Romani Children"

_ijerph, 2021, doi:10.3390/ijerph181910518_

Round 1
Reviewer 1 Report
I really appreciate the effort from the authors of this manuscript: to present an item analysis of the basic subtests of the Czech version of the WJ IV COG battery for a group of Romani children aged 7-11 years in relation to the Czech normative population. There are just few comments which could help to improve the quality of this manuscripts:
- first para of an introduction should tell the readers what is the aim of the manuscript and why they should to read this,
- the references focused on heterogeneous Roma communities and their structured should be mentioned,
- methodology is well described,
- some items in the results are unclear. It would be better to explain them more detailed,
- discussion should be better structured: strengths and limitations, also implications parts are missing
Author Response
Dear reviewer, thank you for your comments. Everything that was requested was added. Then:
The research objective was already included in the original version of the article. This was supplemented with a practical objective.
It was added that the Romani were from the Slovak Romani sub-ethnic group and the description of the population was expanded to include more detailed information.
Most of the items in the article are described, especially the verbal items, but not described are items of a visual and in some cases numerical nature, as these are difficult to operationalize more concisely.

Reviewer 2 Report
General comments
The topic under investigation has a number of omissions in the reporting of the study that need to be addressed. Information is messy and written English is difficult to read. As there are a number of omissions in the design of this study, the value of this topic under review is questionable.
Abstract.
The abstract will need to re-write after finalizing the content of paper.
Methods: general
It may be clearer if methods follow this order “Design”, “Participants”, “Outcome measures”. It may be better to include “Procedure” in the “Design”.
Methods: Design
There are some points missing in this part, for example, the recruitment, the person who collected data and procedure.
Methods: Participants
As this is method part for potential participants, authors just need to provide inclusion and exclusion criteria. As for the actual data, these should be in the result.
Methods: outcome measures
This is missing. No information about questionnaire.
Results: Tables
The format and information are far away the standard of publication, suggesting authors should read other paper or find an example to follow.
Discussion/ Conclusion: context.
As there are so many questions in the method and results, the discussion and conclusion are questionable.
Readability and style.
This paper is not easy to read, because the fluency of written English, suggesting for proofreading.
Author Response
Dear reviewer, thank you for your comments. Everything that was requested was added. Then:
The tables are consistent with the use of a given statistical procedure included in the technical manual of the original US standards. The procedure does not allow for any other variation of the tables.
The discussion was extended.
Proofreading was done by a native speaker.

Reviewer 3 Report
The subject of the article is of great theoretical and especially applied importance. I very much appreciate the addressing of the reliability of the diagnosis of intelligence and other cognitive functions in culturally distinct groups. However, in general, the article is too superficial and needs significant revisions and additions.
Please, provide a rationale for why Roma children were selected as a culturally distinct group. It is worth mentioning what are the population ratios of Roma children in the Czech Republic.
I suggest a detailed characteristics and description of the test structure (scales, subtests) and how each item in the subtests is scored (0 - 1 points, or 0 – 2 or 0-3?).
What do the authors think the difficulties in each item mean? It probably depends on the scoring, but it's not clear. Does only 0 points mean difficulty?
In Results section, it is necessary to provide details of children's scores on each item and whether differences between groups on each item are statistically significant. Only in the second step of the calculation can the dif measure be analyzed and specific values for this indicator should also be given. The phrase "dif measure was lower in.." is too vague.
It is important to answer some substantive and important questions in the article. Were the Roma children's difficulties, if any, due to the fact that they were Roma children or because their families had low socioeconomic status? That is, is cultural difference or SES an important factor? In order to determine this, it is necessary to compare a group of children, for example, from Czech families but with low and high SES with Roma children with low and high SES (if any). I recommend conducting additional research (e.g. involving even if only 30 children) or selecting from the data has been collected, and checking which of these two factors is more important. Especially that the authors write in the theoretical part about the significance of SES for results in cognitive tests.
It is worth informing readers whether a new version of the test for Roma children will be created or has been created based on the results?
One minor note: in Table 1 you need to state/remind which subtests are verbal which are non-verbal.
Author Response
Dear reviewer, thank you for your comments. Everything that was requested was added. Then: The discussion was extended.
Proofreading was done by a native speaker.
Round 2
Reviewer 3 Report
I accept the corrected version of the article.